# Functional Prediction of Microbial Communities in Sediment Microbial Fuel Cells

**DOI:** 10.3390/bioengineering10020199

**Published:** 2023-02-03

**Authors:** Jimmy Kuo, Daniel Liu, Chorng-Horng Lin

**Affiliations:** 1Department of Planning and Research, National Museum of Marine Biology and Aquarium, Pingtung 94450, Taiwan; 2Graduate Institute of Marine Biology, National Dong Hwa University, Pingtung 94450, Taiwan; 3Department of Biomedical Sciences, Da-Yeh University, 168 University Road, Dacun, Changhua 51591, Taiwan

**Keywords:** sediment microbial fuel cell, functional prediction, 16S rDNA, microbial communities

## Abstract

Sediment microbial fuel cells (MFCs) were developed in which the complex substrates present in the sediment could be oxidized by microbes for electron production. In this study, the functional prediction of microbial communities of anode-associated soils in sediment MFCs was investigated based on 16S rRNA genes. Four computational approaches, including BugBase, Functional Annotation of Prokaryotic Taxa (FAPROTAX), the Phylogenetic Investigation of Communities by Reconstruction of Unobserved States (PICRUSt2), and Tax4Fun2, were applied. A total of 67, 9, 37, and 38 functional features were statistically significant. Among these functional groups, the function related to the generation of precursor metabolites and energy was the only one included in all four computational methods, and the sum total of the proportion was 93.54%. The metabolism of cofactor, carrier, and vitamin biosynthesis was included in the three methods, and the sum total of the proportion was 29.94%. The results suggested that the microbial communities usually contribute to energy metabolism, or the metabolism of cofactor, carrier, and vitamin biosynthesis might reveal the functional status in the anode of sediment MFCs.

## 1. Introduction

Microbial fuel cells (MFCs) can grow microorganisms from organic matter supplied by sediments or wastewater to achieve the two goals of bioremediation and energy production [1]. The anodic electron transfer mechanisms can be attributed to the direct transfer via redox-active proteins, direct transfer via conductive pili, indirect transfer via electron shutters, and indirect transfer via reduced metabolites [2]. The exoelectrogenic bacteria *Shewanella oneidensis*, *Geobacter metallireducens*, and *Geobacter sulfurreducens* can use cytochromes as electron transfer proteins or as oxidoreductive enzymes that catalyze the reduction in reactive substrates; such species are responsible for the power produced in MFCs [3,4].

Different sediments might contain different exoelectrogens, and other microorganisms might provide supporting materials, such as oxidized mediators, oxygen-removing species, or nutrients, for the microbial community to adapt to the power output. *Geobacteraceae* [5] and *Desulfobulbaceae* [6] have been shown to predominantly interact in anodic biofilms and were positively correlated with electricity generation efficiency. This syntrophic interaction suggested that *Geobacter* might cooperate with other microorganisms to generate electricity. For example, *Sporomusa* can convert methanol into acetate, which is then utilized by *Geobacter* [7,8]. The electron transfer could be due to quorum-sensing chemicals, minerals as mediators, or cell-cell communication; for instance, the oxidation of methane by anaerobic methanotrophic archaea is linked to sulfate reduction by sulfate-reducing bacteria [3,4]. Methane has been discussed as the substrate for MFCs by the syntrophic association between *Geobacter* and methanotrophs [9]. On the other hand, the power density could be reduced by the presence of nonexoelectrogenic bacteria or nonactive cells that disrupt the electrical conductivity of the biofilm [3]. Knowledge about the metabolic states of bacteria in microbial fuel cells is helpful for a macroscopic understanding of the mechanism of electricity production, but such research is currently lacking.

In this study, we applied functional status prediction using the program BugBase [10], the Functional Annotation of Prokaryotic Taxa (FAPROTAX) [11], Phylogenetic Investigation of Communities by Reconstruction of Unobserved States (PICRUSt2) [12], and Tax4Fun2 [13]. The 16S rDNA sequences derived from different soils of the sediment MFCs were aligned to a databank for taxonomic annotation and then functional prediction. This study aims to provide functional information correlated to the power output in sediment MFCs.

## 2. Materials and Methods

### 2.1. Sediment MFCs and Sequence Processing

The single-chamber, mediator-free sediment MFCs and sequence processing of soil samples D1, D2, D3, D4, D5, S1, S2, S3, S4, and S5 [14] and soil samples D, DA, RF, and RFA [15] were described previously. Briefly, the soil was placed in a transparent plastic container (23 × 14 × 12 cm) with a 5 cm depth, and carbon fiber cloth was buried in the sediment soil as an anode and a carbon rod or a carbon fiber cloth was used under the water level as the cathode. The reason we selected these soil samples is that the soils were all applied as sediment for sediment MFCs. The soil samples Nr1, Nr2, and Nr3 from rice paddy fields of traditional farming were part of the study regarding the soil microbial communities comparison between organic and traditional farming, and the bacterial genomic DNA (gDNA) was extracted from 0.5 g of soil by using a NucleoSpin Soil DNA isolation kit (MACHEREY-NAGEL, Dueren Germany) according to the manufacturer’s instructions, and DNA sequencing was performed at BIOTOOLS Co., Ltd. (New Taipei, Taiwan) (manuscript in preparation). The study information includes sample numbers, soil types, voltage outputs, and groups for program calculation, which are listed in Table 1. The soils used in this study were all collected near the campus in Dacun Township, Changhua County, Taiwan.

### 2.2. Functional Prediction of Microbial Communities

The 16S rDNA sequences were demultiplexed, joined, and denoised, as the clean data were dereplicated with the qiime vsearch dereplicate-sequences command [16]. The derived sequences were considered amplicon sequence variants (ASVs), and the derived table indicated the number of times each ASV was observed in each sample. The table was further filtered with a minimum of 4 reads and 1 sample for a sequence by the qiime feature-table filter-features command to remove the rare sequences for less memory [16]. The filtered sequences were then used at the start of the bioinformatic analyses. Hierarchical clustering was generated in R version 4.0.3 with the scale, dist, and hclust functions [17]. Table 2 lists the sequence references and pathway databases of the four tools.

BugBase is a bioinformatic method for the organism-level coverage of functional pathway prediction [10]. The filtered sequences were subjected to closed-reference clustering at a 97% similarity against the Greengenes 13_8 97% OTUs (operational taxonomic units) reference database [22] with the QIIME vsearch cluster-features-closed-reference command [16] to obtain the OTU table with taxonomy. The table was uploaded to the BugBase website (https://bugbase.cs.umn.edu/upload.html (accessed on 20 April 2022)) for functional prediction. Instead of the default of biologically interpretable trait prediction, the 574 KEGG (Kyoto encyclopedia of genes and genomes) modules [18] were applied for the calculation.

FAPROTAX predicts microbial metabolic functions based on marine culturable microbe functional annotations in a database [11]. The filtered sequences were assigned taxonomy at 99% similarity against the Silva 138 99% OTUs full-length sequences database [21] with the qiime feature-classifier classify-sklearn command [16] for the ASV table with taxonomy. The collapse_table.py script was applied to obtain the function tables.

PICRUSt2 predicts the functions of microbial communities based on marker gene sequencing profiles [12]. The picrust2_pipeline.py command runs the default pipeline with the filtered sequences and table input for pathway analyses. The reference database and pathway database were in the integrated microbial genomes (IMG) [20] and MetaCyc Metabolic Pathway Database [19].

Tax4Fun2 is an R package for the functional prediction of microbial communities from 16S rDNA gene sequences [13]. The default database was applied. First, the runRefBlast command was performed to run the reference blast of Ref99NR with the filtered sequence input, followed by the makeFunctionalPrediction command to predict functional profiles with the filtered table input. KEGG pathways [18] were used as the reference.

### 2.3. Statistical Analysis

The statistical analysis of metagenomic profiles (STAMP) was applied to compare the functional features of each tool between three groups of L (D1, S1, Nr1, Nr2, and Nr3), H1 (D3, D5, S2, S4, S5, and RF), and H2 (D2, D4, S3, D, DA, and RFA) (Table 1) using the Kruskal-Wallis test (*p* < 0.05), followed by the Tukey-Kramer Test for post hoc assay [23].

## 3. Results

### 3.1. Functional Feature Prediction

In this study, 17 soil samples from three independent experiments were grouped into three groups: L, H1, and H2 (Table 1). L represents soils without MFC processing; H1 represents the voltage outputs of the soils larger than 50 mV; and H2 represents the voltage outputs of the soils lower than 50 mV. The voltage output selection of 50 mV is arbitrary. The clean 16S rDNA sequences, after demultiplexing, joining, and denoising was dereplicated as amplicon sequence variants (ASVs), followed by a filtering process with a minimum of four reads and one sample for a sequence. The filtered sequences were assigned taxonomy according to the individual instructions. Appendix A shows the relative abundance of phyla for the soil microbial communities, and except for the unassigned bacteria, the predominant microorganism in soils D1, S2, S3, S4, S5, D, DA, and RF was *Proteobacteria*, whereas in soils D2, D3, D4, D5, and S1 it was *Firmicutes*. In the RFA soil, *Bacteroidota* was the predominant microorganism, whereas *Acidobacteriota* was predominant in Nr1, Nr2, and Nr3 soils. Hierarchical clustering of the soil microbial communities showed that the microbial communities in D, DA, RF and RFA soils were different from the others, and the microbial communities in Nr1, Nr2, and Nr3 soils were further grouped together (Appendix A). The highest three Shannon diversity indexes of the soil microbial communities were 11.248 (Nr2), 11.235 (Nr1), and 11.021 (RF), whereas the lowest three were 8.285 (D3), 8.445 (D2), and 8.57 (RFA) (Appendix A).

The functional features were calculated and followed by statistical analyses (Table 2). There were 67, 9, 37, and 38 functional features shown to be statistically significant from the programs BugBase, FAPROTAX, PICRUSt2, and Tax4Fun2, respectively, and the normalized percentages to the total features were 11.67, 9.78, 8.47, and 10.16% (Table 2). For FAPROTAX, there were 64 functional groups represented with at least one record, and 78.4% of the ASVs were not assigned to any group. In PICRUSt2, the weighted nearest-sequenced taxon index (weighted NSTI) was used to evaluate the average distance for the ASVs in a given sample to a reference bacterial genome, and higher scores (>0.15) might suggest the few related references with low prediction quality [12]. Appendix A lists the weighted NSTI scores of this study, and 8 of the 17 samples had scores higher than 0.15. For Nr1, Nr2, and Nr3, the weighted NSTI scores were 0.52, 0.51, and 0.40, respectively. Tax4Fun2 provides the fraction of taxonomic units that were unused (FTU) and the fraction of sequences unused (FSU) indices as quality indicators [13]. In this study, both the FTU and FSU of all samples were between 0.50 and 0.96, and the indices of Nr1, Nr2, and Nr3 were all larger than 0.90 (Appendix A). These results suggested that the functional predictions were based on a few sequences, and the reason was mainly that the assigned OTUs or ASVs did not match well with the reference databases.

### 3.2. Functional Features Related to Power Generation

According to functional characteristics, we referred to the KEGG pathway maps to classify manually and compare these functional features (Appendix A). Among these features, functions related to energy metabolism were the only functions included in all four calculated programs, and the sum total of the percentage was 93.54%. Subsequently, the functions of the cofactor, carrier, and vitamin biosynthesis, fatty acid, and lipid biosynthesis, and secondary metabolite biosynthesis were included in the three programs, and the sum total percentages were 29.94, 13.65, and 12.16%, respectively (Figure 1). Because we used four different programs with different reference databases and metabolic pathways for the comparison, the mutual function features, such as energy metabolism and metabolism of cofactor, carrier, and vitamin biosynthesis, could suggest that these functions are the majority or the dominant.

Furthermore, we performed a heatmap and clustering analysis with both functional features of energy metabolism and the metabolism of cofactor, carrier, and vitamin biosynthesis for a graphical representation of clustering (Figure 2). Soil samples Nr1, Nr2, and Nr3 were clustered together across all the methods applied, suggesting that similar microbial communities were present, and indeed the soils were collected as replicates from the rice paddy field of traditional farming. However, the clustering results could not reveal the three groups for H1, H2, and L, and it is possible that there were dynamic changes in the microbial communities after the MFC process. We were interested in the functional features that showed higher abundance in the H1 and H2 groups than in the L group because it might suggest that the functions are related to the electron output for microbial communities in sediment MFCs. For example, these functional features were M00124 pyridoxal biosynthesis erythrose 4P pyridoxal 5P and M00126 tetrahydrofuran biosynthesis GTP THF by BugBase (Figure 2A); nitrate reduction, nitrate respiration, nitrogen respiration, methanogenesis by CO_2_ reduction with H_2_, sulfite respiration, and ureolysis by FAPROTAX (Figure 2B); pyridoxal 5′-phosphate biosynthesis I, super pathway of glycolysis, pyruvate dehydrogenase, Tricarboxylic acid (TCA), and glyoxylate bypass, superpathway of menaquinol biosynthesis (menaquinol-6, menaquinol-10, menaquinol-9, demethylmenaquinol-9, and demethylmenaquinol-6), and superpathway of tetrahydrofolate biosynthesis and salvage by PICRUSt2 (Figure 2C); and methane metabolism by Tax4Fun2 (Figure 2D). The exoelectrogens *Aeromonas*, *Bacillus*, *Desulfobulbus*, *Desulfovibrio*, *Enterobacter*, *Geobacter*, *Klesbsiella*, and *Shewanella* were assigned to the functional features related to energy metabolism predicted by FAPROTAX listed in Appendix A. The functional features of metabolism for the generation of precursor metabolites and energy relative to nitrogen, sulfur, methane, and glycolysis metabolism are further presented in bar plots to show the differences in quantity (Figure 3). The functional features of nitrate reduction, nitrate respiration, nitrogen respiration, sulfite respiration, ureolysis, the superpathway of glycolysis, pyruvate dehydrogenase, TCA, glyoxylate bypass, methanogenesis by CO_2_ reduction with H_2_, and methane metabolism in Groups H1 or H2 showed higher quantities than the features in Group L, suggesting that these functions might be helpful for the power output of the microbes in sediment MFCs. Figure 4 presents the bar plots of the functional features of menaquinol, pyridoxal 5′-phosphate, and tetrahydrofolate biosynthesis. All showed higher quantities of features in Groups H1 and H2 than in Group L. The significant post hoc assays by the Tukey-Kramer test are shown in Appendix A. Again, the results strongly suggested that the metabolism of cofactor, carrier, and vitamin biosynthesis was related to power generation.

## 4. Discussion

Functional prediction tools based on 16S rDNA sequences provide an economical and initial resolution for bacterial function and ecological trait annotation in microbial communities and have become popular and widely used. Recently [24] have reported a review regarding the functional prediction from taxonomic genes, including BugBase, FAPROTAX, PICRUSt2, and Tax4Fun2 programs. Briefly, the BugBase tool predicts the functional pathways as well as biologically interpretable traits (Gram staining, oxygen tolerance, biofilm formation, pathogenicity, mobile element content, and oxidative stress tolerance) and is available for use as a web application (http://bugbase.cs.umn.edu (accessed on 20 April 2022)); however, if 16S rDNA sequences were applied, the Greengenes reference database should be used. FAPROTAX prediction is based on the literature of cultured taxa, whereas the main limitation is only the marine prokaryotic organisms are considered, and if the taxonomic resolution is poor, the prediction does not infer the upper rank (e.g., genus). The advantages of PICRUSt2 as evolutionary models are taken into account, and the NTSI confidence score is provided, as well as extensive documentation and active community. The advantages of the Tax4Fun2 are R language which is applied and easily accessible for a large number of users with low experience in bioinformatics, and the confidence scores (FTU and FSU) provided.

The PICRUSt tool has been applied for the investigation of the microbial response to petroleum hydrocarbon contamination and revealed the extrahydrocarbonoclastic activities in contaminated soils [25]. The functional features of soil microbial communities would be affected by long-term tillage practices, and crop rotation combined with no-tillage management could show the highest bacterial diversity and predictive functional capacity [26,27]. Interestingly, predictive functional analysis with PICRUSt suggested that the genes associated with plant fitness and plant growth promotion were abundant in agricultural soil, while the genes related to organic matter degradation were abundant in nonagricultural soil [28]. A study on the dryland soil bacterial community with tax4Fun tools showed that the abundance of the genes involved in nitrogen, carbon, and phosphorous cycles varied among land use systems and seasons [29]. The effects of agricultural management on tomatos [30] and potato plants [31] were revealed by functional predictions of soil microbial communities.

Few functional prediction analyses were applied to the anodic microbial communities of MFCs. A previous study on MFCs applied to anode-enhanced azo degradation demonstrated that MFC processing would enhance manganese-, iron-, fumarate- and nitrate-respiration, soil bioremediation, and chemoheterotrophy but suppress methanogenesis, sulfate respiration, and hydrogen oxidation by FAPROTAX [32]. Studies on the potential of MFCs for antibiotic removal suggested that functional genes related to extracellular electron transfer were increased, but methanogen function genes and multiple antibiotic resistance genes were reduced [33], and functional genes related to metabolism and antibiotic resistance genes were enhanced [34] with the PICRUSt tool. In accordance with the maximum power density and the PICRUSt prediction, the relative abundance of cell mobility, replication, repair, translation, membrane transport, signal transduction, and the metabolism of cofactors and vitamins could be the reason why the electroactive biofilm had high electrocatalysis [35].

We used BugBase, FAPROTAX, PICRUSt2, and Tax4Fun2 tools to predict the functions of microbial communities in sediment MFCs, and energy metabolism and metabolism of cofactor, carrier, and vitamin biosynthesis were the mutual functions with a higher abundance (Figure 1 and Figure 2). Among energy metabolism, there were functional features of nitrate reduction, nitrate respiration, nitrogen respiration, sulfite respiration, ureolysis, the super pathway of glycolysis, pyruvate dehydrogenase, TCA, and glyoxylate bypass, and methane metabolism and methanogenesis by CO_2_ reduction with H_2_ that showed a higher abundance consistent with the power output (Figure 3). Microorganisms involved in the sulfur, nitrate, iron, and methane metabolic pathways might interact with each other [36,37,38,39,40]. The exoelectrogens *Aeromonas*, *Bacillus*, *Desulfobulbus*, *Desulfovibrio*, *Enterobacter*, *Geobacter*, *Klesbsiella*, and *Shewanella* [2]were also assigned to the functional features related to energy metabolism predicted by FAPROTAX (Appendix A). Ureolytic bacteria can utilize urea for nitrogen or energy and affect the composition and morphology of calcium carbonate crystals to enhance biomineralization [41]. The super pathway of glycolysis, pyruvate dehydrogenase, TCA, and glyoxylate bypass integrates several fundamental metabolic reactions for ATP generation [19]. For the metabolism of cofactor, carrier, and vitamin biosynthesis, the biosynthesis of menaquinol, pyridoxal 5′-phosphate, and tetrahydrofolate were positive functions related to the power output (Figure 4). Menaquinones (MK, vitamin K2) are mainly synthesized by bacteria and function as electron carriers in cell membranes, act as antioxidants that protect cell membranes from lipid oxidation, and are involved in the active transport of molecules across cell membranes [19,42]. Pyridoxal 5′-phosphate is a B6 vitamer involved in several metabolic reactions, such as amino acid biosynthesis and degradation, iron metabolism, nucleotide utilization, cofactor biosynthesis, and biofilm formation [43]. Tetrahydrofolate (vitamin B9) might be involved in the metabolism of iron-sulfur clusters by acting as an electron donor [44]. Because the complex microbial communities were applied to the sediment MFCs, these functional features should contribute to and represent the metabolic states in the anode-associated environment.

The possible pitfalls of the functional prediction tools based on 16S rDNA were the limited reference databases, especially to the environmental samples; that is, many of the sequences could not be assigned a taxon for the prediction, and the second concern was the different tools that would perform and derive different functional features [24,45,46]. To improve the limited reference databases, user-defined reference databases could be amended for the analysis [12,13], or multiple tools could be applied for the comparison [46], which was the approach we used in this study. Pyridoxal 5′-phosphate and tetrahydrofolate biosynthesis were predicted by the BugBase and PICRUSt2 tools and showed consistent results. However, for methane metabolism, the functional features M00567 Methanogenesis CO_2_ methane (BugBase), methanogenesis by CO_2_ reduction with H_2_ (FAPROTAX), and methane metabolism (Tax4Fun2) showed different predictions; only the function M00567 Methanogenesis CO_2_ methane showed a higher quantity in Group L than in Groups H1 and H2 (Figure 2). The microbial interactions in anode biofilms appear to be complicated, and dynamics form symbiotic relationships to better adapt to the environment [47]. High throughput sequencing indeed provided microbial analysis at a macro scale. However, to identify soil microorganisms and protein activity during the sampling period for better anodic regulatory mechanism interpretation of sediment MFCs, functional shotgun metagenomics [8] and metatranscriptomics [48] should be the best methods.

## 5. Conclusions

We have demonstrated the functional prediction of microbial communities in sediment MFCs based on 16S rDNA with four tools, BugBase, FAPTROTAX, PICRUSt2, and Tax4Fun2. Both the energy metabolism and the metabolism of cofactor, carrier, and vitamin biosynthesis were the mutual functional features with higher abundance. The metabolic reactions of nitrate reduction, nitrate respiration, nitrogen respiration, sulfite respiration, ureolysis, the super pathway of glycolysis, pyruvate dehydrogenase, TCA, glyoxylate bypass, methane metabolism, methanogenesis by CO_2_ reduction with H_2_, menaquinol biosynthesis, pyridoxal 5′-phosphate biosynthesis, and tetrahydrofolate biosynthesis were predicted to be active in anode-associated soils. If the OTUs or ASVs from soil microbes can be assigned well to the reference databases, this should improve the accuracy and representativeness of the functional prediction.

## Figures and Tables

**Figure 1 bioengineering-10-00199-f001:**
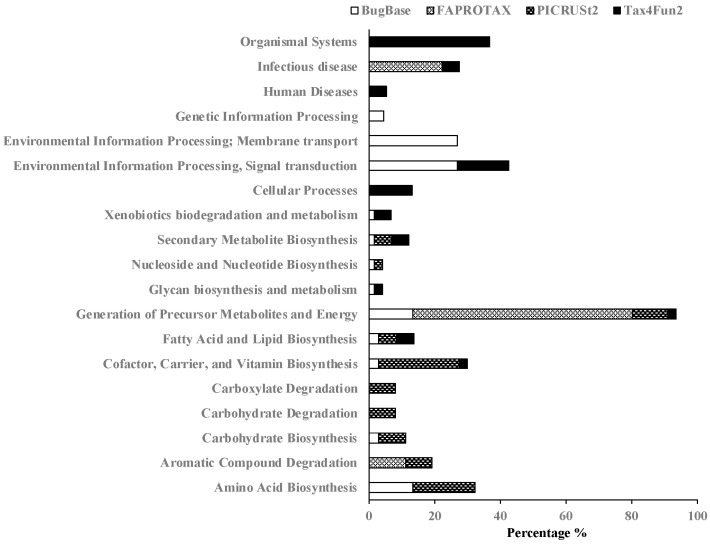
The relative abundance of the pathway categories on the anode-associated soils by the BugBase, FAPTROTAX, PICRUST2, and Tax4Fun2 tools.

**Figure 2 bioengineering-10-00199-f002:**
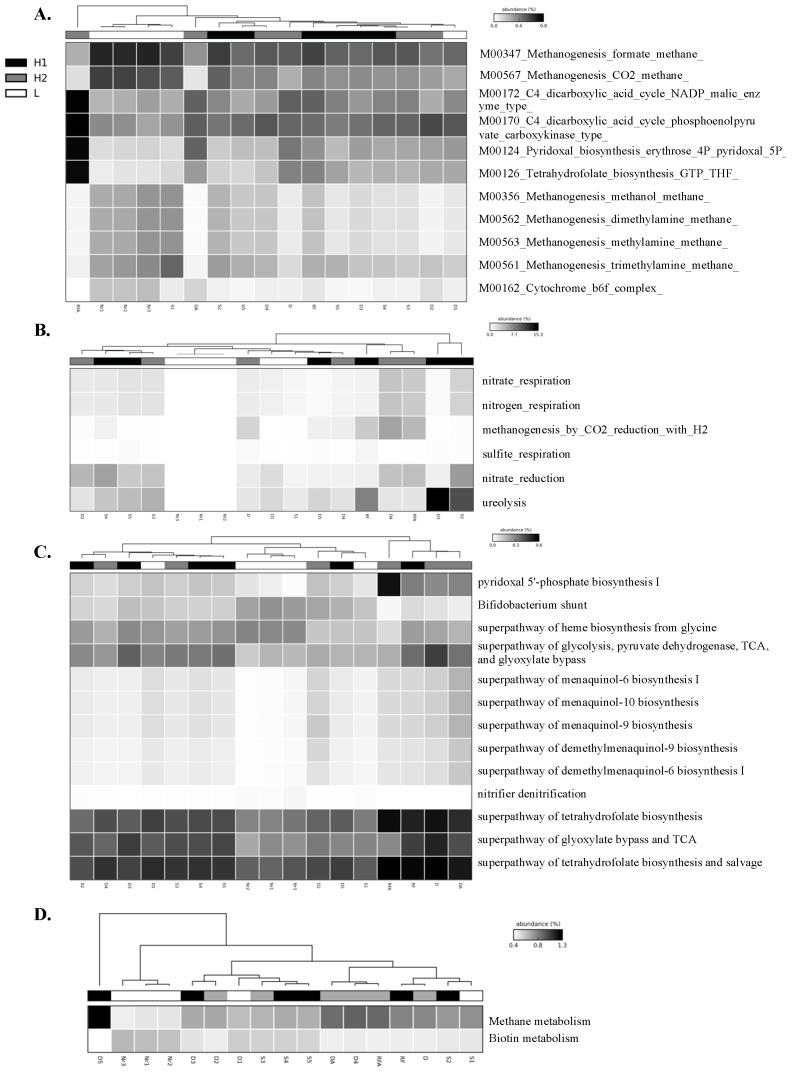
Heatmap of functional features related to energy metabolism and cofactor, carrier, and vitamin biosynthesis metabolism predicted by **A**. BugBase. **B**. FAPROTAX. **C**. PICRUSt2. **D**. Tax4Fun2. The abundance scale bar represents the percentage with grayscale. H1 represents the voltage outputs of soils larger than 50 mV (D3, D5, S2, S4, S5 and RF) with black. H2 represents the voltage outputs of the soils lower than 50 mV (D2, D4, S3, D, DA and RFA) with gray. L represents the soils without MFC processing (D1, S1, Nr1, Nr2 and Nr3) with white.

**Figure 3 bioengineering-10-00199-f003:**
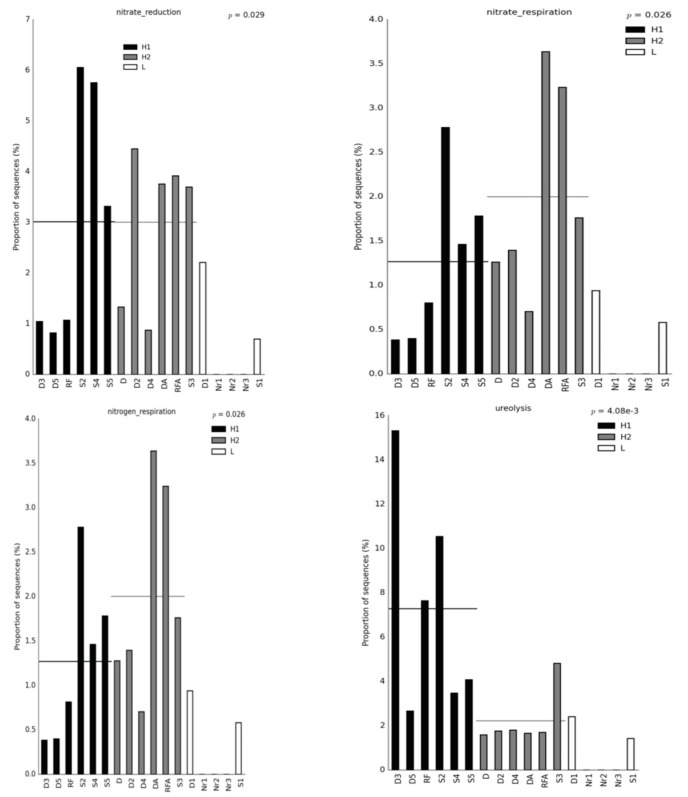
Bar plots of the functional features related to energy metabolism. The x-axis indicates the samples grouped into three groups. H1 represents the voltage outputs of soils larger than 50 mV with black. H2 represents the voltage outputs of the soils lower than 50 mV with gray. L represents the soils without MFCs processing with white. The y-axis indicates the relative abundance. The horizontal line crossing the bar represents the average.

**Figure 4 bioengineering-10-00199-f004:**
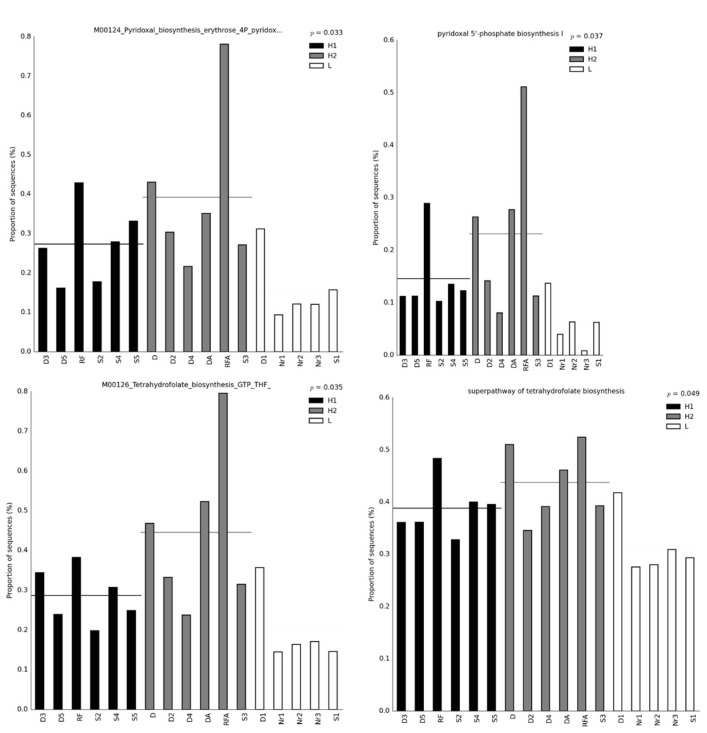
Bar plots of the functional features related to cofactor, carrier, and vitamin biosynthesis. The x-axis indicates the samples grouped into three groups. H1 represents the voltage outputs of soils larger than 50 mV with black. H2 represents the voltage outputs of the soils lower than 50 mV with gray. L represents the soils without MFCs processing with white. The y-axis indicates the relative abundance. The horizontal line crossing the bar represents the average.

**Table 1 bioengineering-10-00199-t001:** Metadata. Study information includes sample numbers, soil types, voltage outputs, and groups for program calculation.

Samples	Soil Type	Location	Voltage, mV	Groups *	References
D1	nonhydric	24°00′ N, 120°36′ E	none	L	[14]
D2	nonhydric		40.92	H2	
D3 **	nonhydric		105.88	H1	
D4	nonhydric		0.2908	H2	
D5	nonhydric		328.89	H1	
S1	hydric	24°00′ N, 120°36′ E	none	L	
S2	hydric		64.605	H1	
S3 **	hydric		16.573	H2	
S4	hydric		153.11	H1	
S5	hydric		179.6	H1	
D	drainage ditch	24°00′ N, 120°33′ E	49	H2	[15]
DA***	drainage ditch		37	H2	
RF	rice paddy field	24°00′ N, 120°34′ E	76	H1	
RFA ***	rice paddy field		22	H2	
Nr1	rice paddy field	23°58′ N, 120°35′ E	None	L	Manuscript in preparation
Nr2	rice paddy field		None	L	
Nr3	rice paddy field		None	L	

* There are three groups based on the soils without MFC processing (L), voltage output larger than 50 mV (H1), and voltage output lower than 50 mV (H2). The voltage output selection of 50 mV is arbitrary. ** Glucose solution (0.1 g/mLs) was applied. *** Soils were previously sterilized for one hour by autoclave.

**Table 2 bioengineering-10-00199-t002:** Overview of four computational programs and the functional features.

Program	Reference Database	Metabolic Pathway Database	Functional Features *
BugBase	Greengenes	KEGG module	67 (574)
FAPROTAX	Silva		9 (92)
PICRUSt2	IMG	MetaCys	37 (440)
Tax4Fun2	Ref99NR (NCBI RefSeq)	KEGG pathway	38 (374)

KEGG: Kyoto encyclopedia of genes and genomes [18]. MetaCyc Metabolic Pathway Database [19]. Integrated microbial genomes (IMG) [20]. Silva ribosomal RNA database [21]. Greengenes databases [22]. * Numbers in brackets indicate the total features in the metabolic pathway database.

## Data Availability

The data and materials are available according to request.

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
