# Peer review of "Functional Prediction of Microbial Communities in Sediment Microbial Fuel Cells"

_bioengineering, 2023, doi:10.3390/bioengineering10020199_

Round 1
Reviewer 1 Report
The authors of the manuscript present a study that is important for understanding the functional characteristics of the microorganisms involved in the MFCs.
Four programs for a comprehensive bioinformatics analysis were used in the work. The results are well presented in tables and figures. The conclusions correspond to the objectives of the study.
It would be good if the authors could explain more clearly why these particular samples were chosen for the study and give some information on environments from where biological communitiea are used as MFC.
Author Response
Response: The statement has been revised as follow: Briefly, the soil was placed in a transparent plastic container (23 x 14 x 12 cm) with 5 cm depth, and carbon fiber cloth was buried in the sediment soil as an anode (-), and a carbon rod or a carbon fiber cloth was used under the water level as the cathode (+). The reason we selected these soil samples is that the soils were all applied as the sediment for sediment MFCs. (line 68-72)
The soils used in this study were all collected near the campus in Dacun Township, Changhua County, Taiwan. (line 80-81)
Reviewer 2 Report
This work applied four computational approaches to compare the functional prediction of microbial communities of anode-associated soils in sediment MFCs based on 16S rRNA genes. To improve further the quality of this paper, the following suggestions are made for the authors' consideration:
1. The author should add more information of samples in Tab.1, Such as operation conditions, species richness, shannon-wiener diversity.
2. Why calculate the cumulative percentages? And how to calculate? The more information should be added in the manuscript.
3. The author should summarize the advantage and disadvantage of four prediction tools, and give suggestion about the applicable conditions of four tools, please add more information in the discussion.
Author Response
- The author should add more information of samples in Tab.1, Such as operation conditions, species richness, shannon-wiener diversity.
Response: The table and statement have been revised as follow:
Briefly, the soil was placed in a transparent plastic container (23 x 14 x 12 cm) with 5 cm depth, and carbon fiber cloth was buried in the sediment soil as an anode (-), and a carbon rod or a carbon fiber cloth was used under the water level as the cathode (+). The reason we selected these soil samples is that the soils were all applied as the sediment for sediment MFCs. (line 68-72)
**Glucose solution (0.1 g/mLs) was applied. (line 88)
***Soils were previously sterilized one hour by autoclave. (line 89)
Hierarchical clustering was generated in R version 4.0.3 with the scale, dist and hclust functions (R 2020). (line 98-99)
Figure S1 showed the relative abundance of phyla of the soil microbial communities, and except the unassigned bacteria, the predominant microorganism in soils D1, S2, S3, S4, S5, D, DA, and RF was Proteobacteria, whereas in soils D2, D3, D4, D5, and S1 was Firmicutes. In RFA soil, the Bacteroidota was the predominant microorganism, whereas Acidobacteriota was the predominant in Nr1, Nr2, and Nr3 soils. Hierarchical clustering of the soil microbial communities showed the microbial communities in D, DA, RF, and RFA soils were different from the others, and the microbial communities in Nr1, Nr2, and Nr3 soils were further grouped together (Figure S2). The highest three Shannon diversity index of the soil microbial communities were 11.248 (Nr2), 11.235 (Nr1), and 11.021 (RF), whereas the lowest three were 8.285 (D3), 8.445 (D2) and 8.57 (RFA) (Table S1). (line 150-160)
- R Core Team (2020) R: A language and environment for statistical computing. R Foundation for Statistical Computing, Vienna, Austria. https://www.r-project.org/ (line 471-472)
- Why calculate the cumulative percentages? And how to calculate? The more information should be added in the manuscript.
Response: We sum the percentages of the same functional feature from each methods, in order to show the relative amount, and the “sum total of” was applied. The statement has been revised as follow:
the sum total of proportion was 93.54% (line 19)
the sum total of proportion was 29.94% (line 20)
and the sum total of percentage was 93.54%. (line 188)
the sum total of (line 190)
- The author should summarize the advantage and disadvantage of four prediction tools, and give suggestion about the applicable conditions of four tools, please add more information in the discussion.
Response: The statement has been revised as follow: Recently Djemiel et al. (2022) have reported a review regarding the functional prediction from taxonomic genes including BugBase, FAPROTAX, PICRUSt2, and Tax4Fun2 programs. Briefly, BugBase tool predicts the functional pathways as well as biologically interpretable traits (Gram staining, oxygen tolerance, biofilm formation, pathogenicity, mobile element content, and oxidative stress tolerance), and is available for use as a web application (http://bugbase.cs.umn.edu), however if 16S rDNA sequences were applied, the Greengenes reference database should be used. FAPROTAX prediction is based on the literature of cultured taxa, whereas the main limitation is only the marine prokaryotic organisms are considered, and if taxonomic resolution is poor, the prediction does not infer upper rank (e.g., genus). The advantages of PICRUSt2 are evolutionary models are taken into account, NTSI confidence score provided, and extensive documentation and active community. The advantages of Tax4Fun2 are R language is applied easily accessible for a large number of users with low experience in bioinformatics, and confidence scores (FTU and FSU) provided. (line 273-286)
Reviewer 3 Report
This manuscript insisted that the functional prediction of microbial communities in sediment microbial fuel cells which investigated based on the 16s rRNA genes. The reviewer thinks this manuscript is well-addressed, and the supported information is reasonable.
However, relatively small sample data is difficult to represent sediment microbial fuel cells. Nonetheless, this manuscript is valuable to understanding the sediment MFCs' microbial function.
The reviewer thinks this manuscript needs to minor improvement in English style and clearly improve figure quality, but the manuscript is qualified for publication in a bioengineering journal.
Minor comment
1. The author mixed the use of the abbreviation of the figure. (fig, or figure)
2. It is difficult to read the x and y-axis in figure 3,4.
3. Figures 3, and 4 look bad and are difficult to recognize.
the important figure is keeping on the figure and non-essential data to replace the supplementary information.
Author Response
Minor comment
- The author mixed the use of the abbreviation of the figure. (fig, or figure)
Response: The word “figure” has been used to replace “fig” in line 150, 157, 191, 201, 211,212, 224, 230, 321, 325, 337, and 365)
- It is difficult to read the x and y-axis in figure 3,4.
Response: The significant post hoc assays by the Tukey‒Kramer test were removed as Figure S3 and S4, thus to improve the resolution of Figure 3 and 4.
- Figures 3, and 4 look bad and are difficult to recognize.
the important figure is keeping on the figure and non-essential data to replace the supplementary information.
Response: The significant post hoc assays by the Tukey‒Kramer test were removed as Figure S3 and S4, thus to improve the resolution of Figure 3 and 4.
Round 2
Reviewer 2 Report
The author has revised the paper accordingly.